# Obstructive sleep apnea increases risk of female infertility: A 14-year nationwide population-based study

Zhu Wei Lim[1‡], I-Duo Wang[2‡], Panchalli Wang[3], Chi-Hsiang Chung ⓘ[4,5], Song-Shan Huang[3], Chien-Chu Huang[6,7], Pei-Yi Tsai[8], Gwo-Jang Wu[9,10]*, Kuo-Hsiang Wu[11]*, Wu-Chien Chien ⓘ[4,5]*

1 Department of Obstetrics and Gynecology, Changhua Christian Hospital, Changhua, Taiwan, 2 Division of Pulmonary and Critical Care Medicine, Department of Internal Medicine, Tri-Service General Hospital, National Defense Medical Center, Taipei, Taiwan, 3 Department of Obstetrics and Gynecology, Chiayi Christian Hospital, Chiayi City, Taiwan, 4 Department of Medical Research, Tri-Service General Hospital, Taipei, Taiwan, 5 School of Public Health, National Defense Medical Center, Taipei, Taiwan, 6 Department of Obstetrics and Gynecology, China Medical University Hospital, Taichung, Taiwan, 7 Graduate Institution of Biomedical Sciences, China Medical University, Taichung, Taiwan, 8 Department of Radiation Oncology, Tri-Service General Hospital, National Defense Medical Center, Taipei, Taiwan, 9 Obstetrics and Gynecology Department, Tri-Service General Hospital, Taipei, Taiwan, 10 Graduate Institute of Medical Sciences, National Defense Medical Center, Taipei, Taiwan, 11 Department of Nursing, Tri-Service General Hospital, National Defense Medical Center, Taipei, Taiwan

‡ Zhu Wei Lim and I-Duo Wang are both the first authors to this manuscript.
* gwojiang@yahoo.com (GJW); kamekame@mail.ndmctsgh.edu.tw (KHW); chienwu@ndmctsgh.edu.tw (WCC)

**Data Availability Statement:** Data are available from the National Health Insurance Research Database (NHIRD) published by Taiwan National Insurance (NHI) Bureau. Under Personal Data

## Abstract

### Objectives

To determine the risk of having OSA in a cohort of female subjects who are infertile and the odds of being infertile in women with OSA.

### Patients and methods

A nationwide, case-control study of female patients 20 years or older diagnosed with female infertility living in Taiwan, from January 1, 2000, through December 31, 2013 (N = 4,078). We identified women who were infertile and created a 2:1 matched control group with women who were not infertile. We used multivariable logistic regression analysis to further estimate the effects of OSA on female infertility.

### Results

In this 14- year retrospective study, we included 4,078 patients having an initial diagnosis of female infertility. Of those women with infertility, 1.38% had a history of OSA compared with 0.63% of fertile controls (p = 0.002). The mean ages in the study groups were 32.19 ± 6.20 years, whereas the mean ages in the control groups were 32.24 ± 6.37years. Women with OSA had 2.101- times the risk of female infertility compared to women without OSA (p<0.001).

Protection Act in Taiwan, personal data cannot be obtained publicly. Interested researchers can sent a formal proposal to the NHIRD (https://dep.mohw.gov.tw/dos/cp-5119-59201-113.html).

**Funding:** This study was supported by Grant Support from Tri-Service General Hospital Research Foundation (TSGH-D-110130, TSGH-B-110012). The funders has no role in study design, data collection and analysis, decision to publish, or preparation of the manuscript.

**Competing interests:** The authors have declared that no competing interests exist.

**Abbreviations:** ACOG, American College of Obstetricians and Gynecologists; COPD, Chronic obstructive pulmonary disease; ICD-9-CM, International Classification of Diseases, Ninth Revision, Clinical Modification; NASD, Non-apnea sleep disorders; NHIRD, National Health Insurance Research Database; OSA, Obstructive sleep apnea; PCOS, Polycystic ovary syndrome.

## Conclusion

Our study showed that OSA is more commonly seen in infertile women and increases the odds that a woman will be infertile. More studies need to be done on the whether or not diagnosing and treating OSA can decrease the rate of infertility.

## Introduction

Obstructive sleep apnea (OSA) is a widely prevalent but often underdiagnosed respiratory disorder characterized by recurrent upper airway obstruction during sleeping. Approximately 5% of the general population experiences OSA; among those aged 30–60 years, the prevalence is 9% in women and 24% in men [1, 2]. The risk of OSA is higher with a family history of OSA, obesity, hypertension, menopause, cigarette smoking, and alcohol consumption [3]. OSA results in sleep fragmentation and repetitive hypoxemia and is associated with various comorbidities, including hypertension, diabetes mellitus, ischemic heart disease, and obesity [4, 5].

According to the American College of Obstetricians and Gynecologists (ACOG), the definition of infertility is the failure to conceive after 1 year or more of regular unprotected sexual intercourse [6]. The prevalence of infertility has increased since 1990; in 2010, approximately 48.5 million individuals worldwide suffered from infertility [7]. Moreover, according to the data published by Taiwan's Ministry of the Interior, the fertility rates of women within child-bearing age decreased from 1.680 births per woman in 2000 to 1.080 births per woman in 2018, with the mean age of the women at their first birth increasing from 22.88 years to 30.90 years during that period. Female infertility can result from various conditions, including endometriosis, pelvic adhesion, polycystic ovary syndrome, tubal blockage, hyperprolactinemia, and congenital or acquired uterine or ovarian abnormalities [8].

Infertility can be associated with multiple factors, such as inflammation, obesity, intermittent hypoxia and sympathetic activation [9, 10]. In our previous study, we have found an increased risk of infertility in female patients with non-apnea sleep disorder [11]. However, little is known about whether OSA is associated with a risk of female infertility.

The aim of the present study was to investigate whether women with OSA had an increased risk of subsequent female infertility. Accordingly, we used Taiwan's National Health Insurance Research Database (NHIRD) for conducting the largest retrospective study to date discussing the association of female infertility with OSA.

## Materials and methods

### Data source

Taiwan launched its National Health Insurance program in 1995. This program covers approximately 99% of the 23.74 million individuals in Taiwan, and 97% of clinics are covered by the system [12]. Taiwan's health care system is insurance-based, and is characterized by its good accessibility, high efficiency, comprehensive population coverage (99% of the 23.74 million residents in Taiwan), relatively low costs and short waiting times [13]. The NHIRD contains health registration records for most of the general population in Taiwan, including details of outpatient, inpatient, and emergency department visits, with diagnoses coded according to the International Classification of Diseases, Ninth Revision, Clinical Modification (ICD-9). The high accuracy and validity of diagnoses in the NHIRD have been demonstrated in previous studies, confirming that the NHIRD offers representative data for medical and health-related research [14–16]. A subset of approximately 1 million patient records from the NHIRD was randomly chosen for inclusion in the Longitudinal Health Insurance Database as

an aid to research projects. The privacy of all individuals registered in the program is ensured via the encryption and conversion of the identification numbers of all records.

The protocol for this study was approved by the Institutional Review Board of the Tri-Service General Hospital (TSGHIRB No. 1-106-05-169).

## Study population

From the outpatient and inpatient data of 989,753 individuals for the period 2000 to 2013 in Taiwan's Longitudinal Health Insurance Database, we identified female patients aged 20–45 years who were diagnosed with infertility. We excluded women who received radiation therapy, chemotherapy, and genital organ surgery (Fig 1). Patients who met the criteria were assigned to the study group. We used two-fold propensity score matching to create a control group by matching each case group member with two other women from the database according to age (by 5-year span) and index date, applying the same exclusion criteria [17, 18]. We compared differences between both groups for the prior exposure of OSA. Both groups were followed until the end of 2013. Our study was conducted by extracting the ICD-9 codes from the NHIRD. The diagnosis of OSA is made by polysomnography and the diagnosis of female infertility is confirmed by the obstetrician and gynecologists. We listed several covariates in our study, including season (Spring, Summer, Autumn, Winter), urbanization level of residence and level of care (medical center, regional, and local hospital). Moreover, common

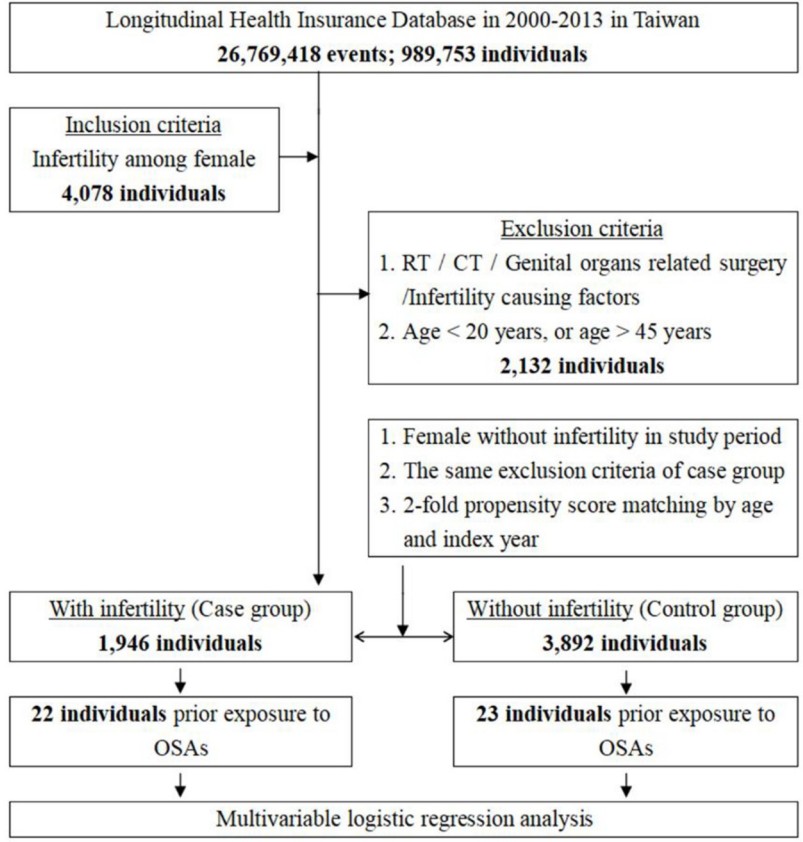

**Fig 1. The flowchart of study design (nested case-control study) from National Health Insurance Research Database in Taiwan.**

comorbidities as well as endocrine and gynecological diseases, such as obesity, Cushing's syndrome, thyroid disease, PCOS, and endometriosis, were used in the analysis to evaluate the cause of infertility and their effects on OSA.

### Statistical analysis

We compared the study and control groups with regard to characteristics and common comorbidities, including hypertension, diabetes mellitus, hyperlipidemia, and COPD, by using chi-squared tests. The mean ages of the two groups were compared using Student's t-test. The odds ratio (OR) for factors potentially associated with female infertility were evaluated using multivariable logistic regression with and without stratification. The variables that adjusted in the odds ratio were the variables that listed in the table. Correlation analysis was used to study the strength of a relationship between age groups and the variables listed in S2 Table. All comparisons were two-tailed, and p-values <0.05 were considered statistically significant. The statistical analyses were performed using IBM SPSS v 22.0 software.

### Results

We identified 4,078 female patients in the database who had been diagnosed with infertility by the end of the study. After applying the exclusion criteria, 2,400 patients were included and assigned as the study group in the analysis. The control group comprised 4,800 matched women without infertility. The mean ages in the study groups were 32.19 ± 6.20 years, whereas the mean ages in the control groups were 32.24 ± 6.37 years. Table 1 summarizes the characteristics of the study and the control groups. There were significantly more patients who had been diagnosed with OSA in the study group than in the control group (1.38% vs. 0.63%; p = .002). In addition, we have found that 33 patients in the study group and 30 patients in the reference group, who have had prior exposure to OSA. Women aged 26–30 years and 31–35 years were more than other aged groups in our study population (28.0% and 32.29% respectively).

Multivariable logistic regression analysis was used in this retrospective study. We listed factors for female infertility in Table 2. There were no significant differences in the risk of female infertility between patients with and without gynecological disorders, endocrine disorders, or concomitant comorbidities, including hypertension, diabetes mellitus, hyperlipidemia, COPD, chronic kidney disease, coronary heart disease, stroke, obesity, anxiety, and depression (Table 2). The increased likelihood of subjects with OSA to be infertile was also showed in Table 2 (adjusted odds ratio, 2.101; p<0.001). ICD-9-CM and correlation analysis were listed in S1 and S2 Tables, respectively.

### Discussion

Our study is the largest retrospective study to date demonstrating the association of female infertility with OSA. In this nationwide, population-based, case-control study of over 14 years, we have found that infertile women have an odds ratio of 2.1 of having OSA compared with women who were fertile but also infertile women were more likely to have OSA. This article is the second study in which our group has shown that women with sleep disorders could link to female infertility. In our previous article, we have found that women with non-apnea sleep disorder had a 3.718-fold risk of female infertility compared with the control cohort [11].

OSA is defined as presence of sleep disordered breathing, excessive daytime sleepiness, and a high apnea–hypopnea index [1, 2, 19]. Changes in the frequency of apneas, hypopneas, and respiratory-effort-related arousal cause increases in sympathetic tone, circadian disruption, systemic inflammation, and intermittent and chronic hypoxia. These factors may gradually

**Table 1. Characteristics of study.**

| Infertility / Variables | Total n | Total % | With n | With % | Without n | Without % | P |
|---|---|---|---|---|---|---|---|
| **Total** | 7,200 | | 2,400 | 33.33 | 4,800 | 66.67 | |
| **OSAs** | | | | | | | 0.002 |
| Without | 7,137 | 99.13 | 2,367 | 98.63 | 4,770 | 99.38 | |
| With | 63 | 0.88 | 33 | 1.38 | 30 | 0.63 | |
| **Age (years)** | 32.22 ± 6.31 | | 32.19 ± 6.20 | | 32.24 ± 6.37 | | 0.751 |
| **Age group (yrs)** | | | | | | | 0.999 |
| 20–25 | 972 | 13.50 | 324 | 13.50 | 648 | 13.50 | |
| 26–30 | 2,016 | 28.00 | 672 | 28.00 | 1,344 | 28.00 | |
| 31–35 | 2,325 | 32.29 | 775 | 32.29 | 1,550 | 32.29 | |
| 36–40 | 1,386 | 19.25 | 462 | 19.25 | 924 | 19.25 | |
| 41–45 | 501 | 6.96 | 167 | 6.96 | 334 | 6.96 | |
| **Insured premium (NT$)** | | | | | | | 0.952 |
| <18,000 | 6,787 | 94.26 | 2,265 | 94.38 | 4,522 | 94.21 | |
| 18,000–34,999 | 272 | 3.78 | 88 | 3.67 | 184 | 3.83 | |
| ≧35,000 | 141 | 1.96 | 47 | 1.96 | 94 | 1.96 | |
| **HTN** | | | | | | | <0.001 |
| Without | 6,751 | 93.76 | 2,204 | 91.83 | 4,547 | 94.73 | |
| With | 449 | 6.24 | 196 | 8.17 | 253 | 5.27 | |
| **DM** | | | | | | | 0.163 |
| Without | 6,850 | 95.14 | 2,271 | 94.63 | 4,579 | 95.40 | |
| With | 350 | 4.86 | 129 | 5.38 | 221 | 4.60 | |
| **Hyperlipidemia** | | | | | | | <0.001 |
| Without | 7,081 | 98.35 | 2,324 | 96.83 | 4,757 | 99.10 | |
| With | 119 | 1.65 | 76 | 3.17 | 43 | 0.90 | |
| **COPD** | | | | | | | 0.003 |
| Without | 7,075 | 98.26 | 2,342 | 97.58 | 4,733 | 98.60 | |
| With | 125 | 1.74 | 58 | 2.42 | 67 | 1.40 | |
| **CKD** | | | | | | | 0.116 |
| Without | 6,993 | 97.13 | 2,342 | 97.58 | 4,651 | 96.90 | |
| With | 207 | 2.88 | 58 | 2.42 | 149 | 3.10 | |
| **IHD** | | | | | | | <0.001 |
| Without | 7,085 | 98.40 | 2,334 | 97.25 | 4,751 | 98.98 | |
| With | 115 | 1.60 | 66 | 2.75 | 49 | 1.02 | |
| **CHD** | | | | | | | 0.018 |
| Without | 7,142 | 99.19 | 2,372 | 98.83 | 4,770 | 99.38 | |
| With | 58 | 0.81 | 28 | 1.17 | 30 | 0.63 | |
| **Stroke** | | | | | | | 0.001 |
| Without | 7,119 | 98.88 | 2,358 | 98.25 | 4,761 | 99.19 | |
| With | 81 | 1.13 | 42 | 1.75 | 39 | 0.81 | |
| **Cancer** | | | | | | | 0.208 |
| Without | 6,989 | 97.07 | 2,321 | 96.71 | 4,668 | 97.25 | |
| With | 211 | 2.93 | 79 | 3.29 | 132 | 2.75 | |
| **Obesity** | | | | | | | 0.001 |
| Without | 7,154 | 99.36 | 2,374 | 98.92 | 4,780 | 99.58 | |
| With | 46 | 0.64 | 26 | 1.08 | 20 | 0.42 | |
| **Hyperestrogenism** | | | | | | | - |

(*Continued*)

**Table 1.** (Continued)

| Infertility Variables | Total n | % | With n | % | Without n | % | P |
|---|---|---|---|---|---|---|---|
| Without | 7,200 | 100.00 | 2,400 | 100.00 | 4,800 | 100.00 | |
| With | 0 | 0.00 | 0 | 0.00 | 0 | 0.00 | |
| **Polycystic ovaries** | | | | | | | 0.479 |
| Without | 7,199 | 99.99 | 2,400 | 100.00 | 4,799 | 99.98 | |
| With | 1 | 0.01 | 0 | 0.00 | 1 | 0.02 | |
| **Irregular menstrual cycle** | | | | | | | <0.001 |
| Without | 7,191 | 99.88 | 2,391 | 99.63 | 4,800 | 100.00 | |
| With | 9 | 0.13 | 9 | 0.38 | 0 | 0.00 | |
| **Endometriosis** | | | | | | | - |
| Without | 7,200 | 100.00 | 2,400 | 100.00 | 4,800 | 100.00 | |
| With | 0 | 0.00 | 0 | 0.00 | 0 | 0.00 | |
| **Uterine leiomyoma** | | | | | | | - |
| Without | 7,200 | 100.00 | 2,400 | 100.00 | 4,800 | 100.00 | |
| With | 0 | 0.00 | 0 | 0.00 | 0 | 0.00 | |
| **Cushing's syndrome** | | | | | | | - |
| Without | 7,200 | 100.00 | 2,400 | 100.00 | 4,800 | 100.00 | |
| With | 0 | 0.00 | 0 | 0.00 | 0 | 0.00 | |
| **Thyrotoxicosis with or without goiter** | | | | | | | 0.139 |
| Without | 7,149 | 99.29 | 2,378 | 99.08 | 4,771 | 99.40 | |
| With | 51 | 0.71 | 22 | 0.92 | 29 | 0.60 | |
| **Acquired hypothyroidism** | | | | | | | 0.056 |
| Without | 7,175 | 99.65 | 2,387 | 99.46 | 4,788 | 99.75 | |
| With | 25 | 0.35 | 13 | 0.54 | 12 | 0.25 | |
| **Anxiety** | | | | | | | <0.001 |
| Without | 6,920 | 96.11 | 2,216 | 92.33 | 4,704 | 98.00 | |
| With | 280 | 3.89 | 184 | 7.67 | 96 | 2.00 | |
| **Depression** | | | | | | | <0.001 |
| Without | 6,974 | 96.86 | 2,262 | 94.25 | 4,712 | 98.17 | |
| With | 226 | 3.14 | 138 | 5.75 | 88 | 1.83 | |
| **Tobacco use disorder** | | | | | | | 0.011 |
| Without | 7,162 | 99.47 | 2,380 | 99.17 | 4,782 | 99.63 | |
| With | 38 | 0.53 | 20 | 0.83 | 18 | 0.38 | |
| **Alcoholism** | | | | | | | 0.019 |
| Without | 7,011 | 97.38 | 2,322 | 96.75 | 4,689 | 97.69 | |
| With | 189 | 2.63 | 78 | 3.25 | 111 | 2.31 | |
| **Season** | | | | | | | 0.720 |
| Spring (Mar-May) | 2,004 | 27.83 | 672 | 28.00 | 1,332 | 27.75 | |
| Summer (Jun-Aug) | 1,892 | 26.28 | 644 | 26.83 | 1,248 | 26.00 | |
| Autumn (Sep-Nov) | 1,581 | 21.96 | 528 | 22.00 | 1,053 | 21.94 | |
| Winter (Dec-Feb) | 1,723 | 23.93 | 556 | 23.17 | 1,167 | 24.31 | |
| **Location** | | | | | | | <0.001 |
| Northern Taiwan | 3,160 | 43.89 | 1,125 | 46.88 | 2,035 | 42.40 | |
| Middle Taiwan | 1,920 | 26.67 | 568 | 23.67 | 1,352 | 28.17 | |
| Southern Taiwan | 1,528 | 21.22 | 533 | 22.21 | 995 | 20.73 | |
| Eastern Taiwan | 411 | 5.71 | 109 | 4.54 | 302 | 6.29 | |
| Outlets islands | 181 | 2.51 | 65 | 2.71 | 116 | 2.42 | |

(Continued)

**Table 1.** (Continued)

| Infertility | Total | | With | | Without | | P |
|---|---|---|---|---|---|---|---|
| Variables | n | % | n | % | n | % | |
| **Urbanization level** | | | | | | | <0.001 |
| 1 (The highest) | 3,068 | 42.61 | 1,086 | 45.25 | 1,982 | 41.29 | |
| 2 | 2,541 | 35.29 | 863 | 35.96 | 1,678 | 34.96 | |
| 3 | 622 | 8.64 | 197 | 8.21 | 425 | 8.85 | |
| 4 (The lowest) | 969 | 13.46 | 254 | 10.58 | 715 | 14.90 | |
| **Level of care** | | | | | | | <0.001 |
| Hospital center | 2,433 | 33.79 | 1,077 | 44.88 | 1,356 | 28.25 | |
| Regional hospital | 2,170 | 30.14 | 621 | 25.88 | 1,549 | 32.27 | |
| Local hospital | 2,597 | 36.07 | 702 | 29.25 | 1,895 | 39.48 | |

P: Chi-square / Fisher exact test on category variables and t-test on continue variables

increase oxidative stress, leading to infertility [20]. Lifestyle factors such as obesity and reproductive pathological and physiological factors such as PCOS, endometriosis, and pregnancy can result in the generation of reactive oxygen species by suppressing the production of NO, promoting the production of endothelium-derived vasoconstrictors, and increasing hemoglobin-mediated inactivation [21, 22]. Therefore, excessive oxidative stress can result in poor oocyte quality, abnormal fertilization, the impairment of blastocyst or embryo development and growth, and increased embryo fragmentation, leading to apoptosis and ultimately leading to failed implantation [23]. It has also been reported that an imbalance of pro-inflammatory cytokines, anti-inflammatory cytokines, chemokines, growth factors, and anti-apoptotic proteins were associated with infertility and unsuccessful in vivo fertilization [24–26].

Patients with OSA are at risk of metabolic disorders, including an irregular menstrual cycle and obesity [27]. The menstrual cycle is modulated by the hypothalamus–pituitary–gonadal axis. The hypothalamus releases gonadotropin-releasing hormone, the pituitary gland produces follicle-stimulating hormone and luteinizing hormone, and the ovary produces estrogen and testosterone. Kloss et al. hypothesized that sleep disorders may activate the hypothalamus–pituitary–gonadal axis and alter sex hormones in the follicular, ovulation, luteal, and menstruation phases, ultimately resulting in infertility [28]. In addition, the interruption of breathing during sleep as a result of OSA can cause circadian dysrhythmia due to the increased levels of melatonin and cortisol [29]. The increase of cortisol levels can downregulate the hypothalamic-pituitary-adrenal axis and inhibit GnRH at the pituitary level, which may alter sex hormone profiles and thus lead to infertility [30].

Scientists have demonstrated the prevalence of OSA in women is lower than in men [2]. There is also likely underdiagnosis of OSA in women due to atypical symptoms such as depression, headache, anxiety and insomnia are more frequent presented in women [31]. In addition, women with OSA have been shown to have greater hypoxic chemosensitivity than those without OSA [32]. In contrast, no central compensatory respiratory drive adaptation has been observed in obese men with or without OSA [32]. The sex hormones progesterone and estrogen have been reported to increase ventilatory control and hypercapnic and hypoxia chemosensitivity [33, 34]. In addition, sex hormone also participated in the distribution of adipose tissue and muscle function in upper respiratory tract [35]. Scientist have demonstrated that high progesterone level in pregnancy status had the protective role in developing OSA, even in obese pregnant women [36, 37]. Greater activity of dilator muscle in upper respiratory tract was also seen in progesterone-dominant luteal phase [38]. However, these effects decreased

**Table 2. Factors of infertility by using multivariable conditional logistic regression.**

| Variables | Crude OR | 95% CI | 95% CI | P | Adjusted OR | 95% CI | 95% CI | P |
|---|---|---|---|---|---|---|---|---|
| **OSAs** | Reference | | | | Reference | | | |
| | 1.900 | 1.056 | 3.417 | 0.001 | 2.101 | 1.118 | 3.950 | <0.001 |
| **Age group** | | | | | | | | |
| 20–25 | 1.000 | 0.818 | 1.193 | 0.999 | 0.947 | 0.772 | 1.160 | 0.753 |
| 26–30 | 1.000 | 0.821 | 1.188 | 0.999 | 0.852 | 0.697 | 1.040 | 0.179 |
| 31–35 | 1.000 | 0.804 | 1.213 | 0.999 | 0.781 | 0.626 | 0.989 | 0.050 |
| 36–40 | 1.000 | 0.739 | 1.320 | 0.999 | 0.783 | 0.572 | 1.071 | 0.165 |
| 41–45 | Reference | | | | Reference | | | |
| **Insured premium** | | | | | | | | |
| <18,000 | Reference | | | | Reference | | | |
| 18,000–34,999 | 0.858 | 0.544 | 1.352 | 0.536 | 1.047 | 0.640 | 1.712 | 0.759 |
| ≧35,000 | 0.861 | 0.459 | 1.618 | 0.663 | 0.782 | 0.396 | 1.543 | 0.511 |
| **HTN** | Reference | | | | Reference | | | |
| | 1.860 | 0.537 | 2.379 | 0.532 | 1.745 | 0.450 | 2.264 | 0.315 |
| **DM** | Reference | | | | Reference | | | |
| | 1.932 | 0.593 | 2.465 | 0.761 | 1.874 | 0.534 | 2.460 | 0.668 |
| **Hyperlipidemia** | Reference | | | | Reference | | | |
| | 1.988 | 0.478 | 3.041 | 0.987 | 2.030 | 0.466 | 3.319 | 0.850 |
| **COPD** | Reference | | | | Reference | | | |
| | 1.598 | 0.328 | 2.091 | 0.101 | 1.593 | 0.320 | 2.133 | 0.139 |
| **CKD** | Reference | | | | Reference | | | |
| | 1.282 | 0.035 | 3.292 | 0.238 | 1.193 | 0.024 | 2.743 | 0.151 |
| **IHD** | Reference | | | | Reference | | | |
| | 1.517 | 0.257 | 2.041 | 0.069 | 1.393 | 1.184 | 1.826 | 0.017 |
| **CHD** | Reference | | | | Reference | | | |
| | 1.538 | 0.150 | 2.931 | 0.347 | 1.397 | 0.107 | 2.558 | 0.201 |
| **Stroke** | Reference | | | | Reference | | | |
| | 1.564 | 0.117 | 3.717 | 0.479 | 1.449 | 0.087 | 3.442 | 0.371 |
| **Cancer** | Reference | | | | Reference | | | |
| | 1.231 | 1.120 | 1.447 | <0.001 | 1.125 | 1.058 | 1.256 | <0.001 |
| **Obesity** | Reference | | | | Reference | | | |
| | 1.705 | 0.253 | 2.959 | 0.511 | 1.744 | 0.258 | 3.206 | 0.627 |
| **Hyperestrogenism** | Reference | | | | Reference | | | |
| | - | - | - | - | - | - | - | - |
| **Polycystic ovaries** | Reference | | | | Reference | | | |
| | - | - | - | - | - | - | - | - |
| **Irregular menstrual cycle** | Reference | | | | Reference | | | |
| | - | - | - | - | - | - | - | - |
| **Endometriosis** | Reference | | | | Reference | | | |
| | - | - | - | - | - | - | - | - |
| **Uterine leiomyoma** | Reference | | | | Reference | | | |
| | - | - | - | - | - | - | - | - |
| **Cushing's syndrome** | Reference | | | | Reference | | | |
| | - | - | - | - | - | - | - | - |
| **Thyrotoxicosis with or without goiter** | Reference | | | | Reference | | | |
| | 2.414 | 0.727 | 3.749 | 0.286 | 1.877 | 0.437 | 2.796 | 0.778 |

(*Continued*)

**Table 2.** (Continued)

| Variables | Crude OR | 95% CI | 95% CI | *P* | Adjusted OR | 95% CI | 95% CI | *P* |
|---|---|---|---|---|---|---|---|---|
| **OSAs** | **Reference** | | | | **Reference** | | | |
| **Acquired hypothyroidism** | Reference | | | | Reference | | | |
| | 2.098 | 0.367 | 4.280 | 0.840 | 1.917 | 0.281 | 4.061 | 0.914 |
| **Anxiety** | Reference | | | | Reference | | | |
| | 1.759 | 0.270 | 3.132 | 0.610 | 1.879 | 0.272 | 3.904 | 0.859 |
| **Depression** | Reference | | | | Reference | | | |
| | 1.600 | 0.257 | 2.400 | 0.246 | 1.644 | 0.253 | 2.689 | 0.404 |
| **Tobacco use disorder** | Reference | | | | Reference | | | |
| | 1.852 | 0.896 | 2.571 | 0.184 | 1.829 | 0.885 | 2.540 | 0.182 |
| **Alcoholism** | Reference | | | | Reference | | | |
| | 1.532 | 0.722 | 2.386 | 0.223 | 1.513 | 0.714 | 2.357 | 0.220 |
| **Season** | | | | | | | | |
| Spring | Reference | | | | Reference | | | |
| Summer | 1.018 | 0.875 | 1.186 | 0.684 | 0.980 | 0.831 | 1.154 | 0.937 |
| Autumn | 0.960 | 0.824 | 1.118 | 0.702 | 0.938 | 0.796 | 1.105 | 0.622 |
| Winter | 0.871 | 0.746 | 1.017 | 0.110 | 0.823 | 0.696 | 0.972 | 0.044 |
| **Location** | | | | | | | | |
| Northern Taiwan | Reference | | | | **Had multicollinearity with urbanization level** | | | |
| Middle Taiwan | 0.794 | 0.696 | 0.906 | 0.001 | **Had multicollinearity with urbanization level** | | | |
| Southern Taiwan | 0.865 | 0.751 | 0.996 | 0.045 | **Had multicollinearity with urbanization level** | | | |
| Eastern Taiwan | 0.871 | 0.659 | 1.153 | 0.377 | **Had multicollinearity with urbanization level** | | | |
| Outlets islands | 0.899 | 0.419 | 1.929 | 0.799 | **Had multicollinearity with urbanization level** | | | |
| **Urbanization level** | | | | | | | | |
| 1 (The highest) | 3.459 | 2.808 | 4.262 | <0.001 | 1.639 | 1.304 | 2.060 | <0.001 |
| 2 | 2.003 | 1.624 | 2.469 | <0.001 | 1.182 | 0.947 | 1.475 | 0.087 |
| 3 | 0.900 | 0.665 | 1.218 | 0.539 | 0.907 | 0.667 | 1.235 | 0.629 |
| 4 (The lowest) | Reference | | | | Reference | | | |
| **Level of care** | | | | | | | | |
| Hospital center | 6.324 | 5.418 | 7.381 | <0.001 | 5.734 | 4.841 | 6.791 | <0.001 |
| Regional hospital | 2.272 | 1.939 | 2.662 | <0.001 | 2.322 | 1.974 | 2.730 | <0.001 |
| Local hospital | Reference | | | | Reference | | | |

OR = odds ratio, CI = confidence interval, Adjusted OR: Adjusted variables listed in the table

with age because of the higher incidence rate of OSA and the greater severity of OSA in post-menopausal women than in premenopausal women [33, 39]. It has been reported that estradiol withdrawal was associated with a predisposition to OSA in peri- and postmenopausal women with depression [40]. As a result, sex hormones may influence the severity of OSA, especially in younger women who desired to get pregnant [41].

This study had some limitations. First, although the diagnosis of OSA is made by polysomnography, we could not obtain the severity of OSA due to the de-identification in the database. Therefore, studies regarding the severity of OSA and subsequent female infertility are warranted. Second, although the study draws subjects from a large database, however, the number of infertility and OSA was only 33 and the number of OSA with no infertility in the control group was only 30 subjects. Third, the absence of women with endometriosis and PCOS in the infertile cohort potentially limits the applicability in populations where these disorders are

more common. Despite the listed limitations, our study provided a large group of patients and its longitudinal effects of over 14 years.

## Conclusion

In conclusion, this study showed that OSA is more commonly seen in infertile women and increases the odds that a woman will be infertile. Therefore, infertile women should be screened for signs and symptoms of OSA, which may help to increase female fertility rate.

## Supporting information

**S1 Table. Abbreviation and ICD-9-CM.**
(DOCX)

**S2 Table. Correlation between variables listed in the table and age group.**
(DOCX)

## Acknowledgments

We appreciate the Health and Welfare Data Science Center, Ministry of Health and Welfare (HWDC, MOHW), Taiwan, for providing the National Health Insurance Research Database (NHIRD).

## Author Contributions

**Conceptualization:** I-Duo Wang, Gwo-Jang Wu.

**Data curation:** Chi-Hsiang Chung.

**Formal analysis:** Zhu Wei Lim, Panchalli Wang, Chi-Hsiang Chung.

**Funding acquisition:** Kuo-Hsiang Wu, Wu-Chien Chien.

**Investigation:** Panchalli Wang.

**Methodology:** Chi-Hsiang Chung, Song-Shan Huang.

**Project administration:** Chien-Chu Huang, Gwo-Jang Wu, Kuo-Hsiang Wu.

**Resources:** Song-Shan Huang, Chien-Chu Huang.

**Software:** Song-Shan Huang, Chien-Chu Huang, Pei-Yi Tsai.

**Supervision:** Gwo-Jang Wu, Wu-Chien Chien.

**Validation:** Chien-Chu Huang, Pei-Yi Tsai.

**Visualization:** Chien-Chu Huang, Pei-Yi Tsai.

**Writing – original draft:** Zhu Wei Lim, I-Duo Wang.

**Writing – review & editing:** Zhu Wei Lim, I-Duo Wang, Wu-Chien Chien.

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
