## [Decision Letter · Decision Letter 0]

23 Jun 2021

PONE-D-21-15342

Obstructive sleep apnea increases risk of female infertility : A 14-year follow-up nationwide population-based cohort study

PLOS ONE

Dear Dr. Chien,

Thank you for submitting your manuscript to PLOS ONE. After careful consideration, we feel that it has merit but does not fully meet PLOS ONE’s publication criteria as it currently stands. Therefore, we invite you to submit a revised version of the manuscript that addresses the points raised during the review process.

The authors have serious and reasonable comments about this manuscript.  The most compelling is the study design itself; this editor agrees that what the authors have shown is that amongst infertile women, OSA is more common, not that OSA is a potential cause of infertility.  

In addition, the editor would like to know how the diagnoses were searched for in the database. Were ICD codes used?  If so, the authors need to indicate which codes were searched for (this could be done in a table).  

We look forward to receiving your revised manuscript.

Kind regards,

James Andrew Rowley

Academic Editor

PLOS ONE

Journal Requirements:

2. Please amend either the abstract on the online submission form (via Edit Submission) or the abstract in the manuscript so that they are identical.

Additional Editor Comments (if provided):

Reviewers' comments:

Reviewer's Responses to Questions

**Comments to the Author**

1. Is the manuscript technically sound, and do the data support the conclusions?

Reviewer #1: Partly

Reviewer #2: No

2. Has the statistical analysis been performed appropriately and rigorously? 

Reviewer #1: I Don't Know

Reviewer #2: I Don't Know

3. Have the authors made all data underlying the findings in their manuscript fully available?

Reviewer #1: Yes

Reviewer #2: Yes

4. Is the manuscript presented in an intelligible fashion and written in standard English?

Reviewer #1: Yes

Reviewer #2: No

5. Review Comments to the Author

Reviewer #1: This population-based cohort study evaluated whether women with OSA had increased risk of subsequent infertility. This study explores an interesting topic, but there are some significant issues noted below.

Major points:

1. This study draws subjects from a large database, however, the number of with infertility and OSA was only 33 with only 30 subjects with OSA and no infertility in the control group.

2. It is not clear what prior exposure to OSA means. Is this the presence of disease at the start of the study, the development of OSA during the time of the study, or both? Similarly, it’s not clear what the “years of OSA exposure” is referring to. I’m not sure what Table 4 is trying to tell me.

3. The groups were then stratified into five-year age groups, making the numbers even smaller and the conclusions less robust.

4. The definition of infertility includes a time period of at least one year. How would the authors know which season has the highest risk for infertility?

5. The limitations section is very brief and should address the issue of the small sample sizes of the OSA/infertility and OSA/non-infertility groups.

6. The authors conclude that detection and diagnosis of OSA may help to increase fertility, but this is speculative. There is no clear data to support this.

Minor points:

1. Many acronyms are not defined when first used: NHIRD, NASD, PCOS, COPD.

2. The last sentence of the “Study Population” section is incomplete.

3. Under “Results,” a sentence reads “Women aged 26-30 years… had the highest percentage.” The highest percentage of what?

4. Reference 21 in the manuscript does not appear to refer to reference 21 in the References section.

5. in the discussion section, the word choice of “elderly” is not appropriate, since elderly women are clearly post-menopausal and not candidates for infertility.

6. There is a sentence in the discussion that reads, The increased of cortisol…” A word appears to be missing.

7. Under the Results section, the Tables are mislabeled.

Reviewer #2: This is a study of the impact of OSA on fertility in women. The authors did a retrospective nested 2:1 matched control study of women with and without infertility and analyzed OSA as a risk factor for infertility, accounting for multiple co-morbidities. This is a companion to their prior prospective study on non-apnea sleep disorders (NASD) and the relationship to infertility. The authors demonstrate an association of OSA with female infertility with an odds ratio of 2.1 for women with infertility to have OSA versus those without infertility and also conclude that length of exposure to OSA increases risk of infertility. However, they conclude that women with OSA are more likely to be infertile than those without, which was not the design of their study.

Strengths: The relationship between OSA and fertility is an important topic and they used a large sample size and they performed sophisticated analyses on many co-variates.

MAJOR WEAKNESSES

General:

1. There is confusion as to what the authors are trying to prove. They conclude that women with OSA are more likely to be infertile but what they show is that women who are infertile have greater odds of having OSA than women who are fertile. This study is not the same design as the prior work, where they looked at women with NASD and then did a prospective study to see the risk of infertility.

2. They do not mention how they define infertility nor if they excluded male infertility.

3. It is surprising that a cohort of 2400 women with infertility has no cases of endometriosis (with an estimated prevalence of 25-50% among infertile women) or PCOS (present in about 40% of women with infertility) making generalizability of any conclusions questionable.

4. There is no discussion of how they determined the duration of the diagnosis of OSA relative to the timing of the diagnosis of infertility.

5. The abstract suggests a different protocol, which would have been to look at women with OSA and see how many are infertile while the actual protocol was to determine odds ratio of having OSA in a cohort of subjects who are infertile.

Introduction:

6. There is a discussion about effect of physiological consequences of OSA on sperm production in mice. This does not seem relevant to a paper on female infertility. Consider discussing more about female-relevant hormones, such as you do in the discussion.

Study Population:

7. Did you exclude women who were infertile because of male infertility issues?

8. How did you determine the prior exposure of OSA?

9. Last sentence is incomplete.

Statistical Analysis

10. There is mention of a Table 5 that is not included in the submission.

Results:

11. Table 3 is confusing—consider deleting these data. Table 2 is sufficient

Discussion:

12. You did not demonstrate that women with OSA had twice the risk of infertility but, rather, that infertile women had an odds ratio of 2.1 of having OSA compared with women who were fertile.

13. The fifth paragraph is confusing and the conclusion does not follow. The sex hormones do not mask the severity—they influence the severity.

MINOR WEAKNESSES

Suggested Content Edits:

Study Population:

1. “The diagnosis of OSA is made BY polysomnography . . . .”

Statistical Analysis:

2. “Correlation[s] analysis WAS used”

3. “Women aged 26-30 years and 31-35 years had the highest percentage”. I believe you mean “constituted the highest percentage”. If this is not what you mean, then please state what they had the highest percentage of.

Discussion,

4. 1st paragraph, last sentence: “This article is the second study that our groups have shown that women . . . infertility”. Consider substituting instead “This article is the second study in which our group has shown that women with sleep disorders . . . infertility.”

5. 3rd paragraph: “. . . elderly women are more likely to become infertile”—I believe you want to say “. . . OLDER women are more likely to BE infertile”.

Endnotes:

6. Endnotes 31 and 32 do not relate to the sentence they are in.

6. PLOS authors have the option to publish the peer review history of their article (what does this mean?). If published, this will include your full peer review and any attached files.

Reviewer #1: No

Reviewer #2: No

---

## [Author Response · Author response to Decision Letter 0]

27 Jul 2021

Answer to Editor’s and Reviewer's comments

Thank you for your positive comments on this manuscript. The responses to the raised questions are below.

The authors have serious and reasonable comments about this manuscript. The most compelling is the study design itself; this editor agrees that what the authors have shown is that amongst infertile women, OSA is more common, not that OSA is a potential cause of infertility. In addition, the editor would like to know how the diagnoses were searched for in the database. Were ICD codes used? If so, the authors need to indicate which codes were searched for (this could be done in a table). 

Response: Thank you for your constructive critique. We totally agreed with you. OSA is more common, not that OSA is a potential cause of infertility. Due to it is a case control study, therefore, association of infertility and OSA was established in these 14 years follow up retrospective population base study. We used ICD-9-CM and added table S1 to indicate the codes. 

Reviewer #1: 

Major points:

1. This study draws subjects from a large database, however, the number of with infertility and OSA was only 33 with only 30 subjects with OSA and no infertility in the control group.

Response: Thank you for your thorough review and salient observations. The limited number of subjects in our study is because we have excluded women who received radiation therapy, chemotherapy, genital organ surgery, and aged <20, >45 years old. We identified women who became infertile and created a 2:1 matched control group with women who were not infertile, and got 4,800 of fertility control and 2,400 of infertility case. After that, we traced back to their OSA exposure, with 33 subjects in the case group and 30 subjects in the control group. 

2. It is not clear what prior exposure to OSA means. Is this the presence of disease at the start of the study, the development of OSA during the time of the study, or both? Similarly, it’s not clear what the “years of OSA exposure” is referring to. I’m not sure what Table 4 is trying to tell me.

Response: Thank you for your thorough review and salient observations. Our study is a case control study which firstly identified infertile women, then traced back to their history of OSA. What table 4 (now table 3) is trying to say is that the more years of having OSA, the higher the risk of having infertility.

3. The groups were then stratified into five-year age groups, making the numbers even smaller and the conclusions less robust.

Response: Thank you for your thorough review and salient observations. 

Although the number is lesser in stratified groups, we have the largest participants (N=989,753) from NHIRD database, which has not been previously reported. And we have adjusted the odd ratio by multivariable conditional logistic regression, which we think that the p value of each stratified variables was representative. The reason why we stratified into five-year age groups is because we wanted to dig deeper into the age-related effect on OSA and infertility and provide a greater precision, rather than the entire age group. 

4. The definition of infertility includes a time period of at least one year. How would the authors know which season has the highest risk for infertility?

Response: Thank you for your thorough review and salient observations. 

This is a 14-year follow-up case control study, first identified the infertility then retrospectively found out previous exposure of OSA, therefore, infertility was definitely diagnosed after a time period of at least one year. We also divided 365 days into four seasons for the first time of infertility diagnosis. 

5. The limitations section is very brief and should address the issue of the small sample sizes of the OSA/infertility and OSA/non-infertility groups.

Response: Thank you for your thorough review and salient observations. We have revised these mistakes based on your recommendation in limitations section. 

Third, the study has the small sample size after retrospectively extracted previous OSA exposure from participants with infertility and no infertility, however, the study has the largest database (N=989,753), which has not been previously reported. 

6. The authors conclude that detection and diagnosis of OSA may help to increase fertility, but this is speculative. There is no clear data to support this.

Response: Thank you for your thorough review and kind suggestions. Indeed, there is no clear data to date about the association between OSA and female infertility. Therefore, we have performed this retrospective study to proof this hypothesis. In our study, we have found increased risk of female infertility in OSA patients. However, as this is a case control study, we could only determine the association between infertility and OSA. Therefore, we revised these mistakes based on your recommendation in Conclusion part. 

In conclusion, this study showed OSA is more commonly seen in infertile women,

especially those with concomitant comorbidities.

Minor points:

1. Many acronyms are not defined when first used: NHIRD, NASD, PCOS, COPD.

Response: Thank you for your thorough review and salient observations. We have revised these mistakes based on your recommendation by adding abbreviation list. 

Abbreviation list:

NHIRD, National Health Insurance Research Database

NASD, Non-apnea sleep disorders

PCOS, Polycystic ovary syndrome

COPD, Chronic obstructive pulmonary disease

OSA, Obstructive sleep apnea

ICD-9-CM, International Classification of Diseases, Ninth Revision, Clinical Modification

ACOG, American College of Obstetricians and Gynecologists

2. The last sentence of the “Study Population” section is incomplete.

Response: Thank you for your thorough review and salient observations. We have revised these mistakes based on your recommendation 

Moreover, common comorbidities as well as endocrine and gynecological diseases, such as obesity, Cushing syndrome, thyroid disease, PCOS, and endometriosis, were used in the analysis to evaluate the cause of infertility and their effects on OSA. 

3. Under “Results,” a sentence reads “Women aged 26-30 years… had the highest percentage.” The highest percentage of what?

Response: Thank you for your thorough review and salient observations. We have revised these mistakes based on your recommendation. 

Women aged 26-30 years and 31-35 years were more than other aged groups in our study population (28.0% and 32.29% respectively).

4. Reference 21 in the manuscript does not appear to refer to reference 21 in the References section.

Response: Thank you for your thorough review and salient observations. We have revised these mistakes based on your recommendation. 

We have deleted reference 21.

5. in the discussion section, the word choice of “elderly” is not appropriate, since elderly women are clearly post-menopausal and not candidates for infertility.

Response: Thank you for your thorough review and salient observations. We have revised these mistakes based on your recommendation. 

In the general population, older women are more likely to be infertile than younger women.

6. There is a sentence in the discussion that reads, The increased of cortisol…” A word appears to be missing.

Response: Thank you for your thorough review and salient observations. We have revised these mistakes based on your recommendation. 

The increased of cortisol levels downregulated the hypothalamic-pituitary-adrenal axis and involved in the inhibition of GnRH at the pituitary level, which altered sex hormone profile and thus leading to infertility.

7. Under the Results section, the Tables are mislabeled.

Response: Thank you for your thorough review and salient observations. We have revised these mistakes based on your recommendation (page 10-11). 

 

Reviewer #2: 

MAJOR WEAKNESSES

General:

1. There is confusion as to what the authors are trying to prove. They conclude that women with OSA are more likely to be infertile but what they show is that women who are infertile have greater odds of having OSA than women who are fertile. This study is not the same design as the prior work, where they looked at women with NASD and then did a prospective study to see the risk of infertility.

Response: Thank you for your thorough review and salient observations. 

Our previous study was a prospective study but present study is a case-control study.

2. They do not mention how they define infertility nor if they excluded male infertility.

Response: Thank you for your thorough review and salient observations. We have revised these mistakes based on your recommendation in introduction section.

(1) According to the American College of Obstetricians and Gynecologists (ACOG), the definition of infertility is the failure to conceive after 1 year or more of regular unprotected sexual intercourse, https://www.acog.org/womens-health/faqs/treating-infertility. 

(2) We extracted data from female infertility which was definitely diagnosed in these 14 years follow up case control study after male infertility has been excluded. 

3. It is surprising that a cohort of 2400 women with infertility has no cases of endometriosis (with an estimated prevalence of 25-50% among infertile women) or PCOS (present in about 40% of women with infertility) making generalizability of any conclusions questionable.

Response: Thank you for your thorough review and salient observations. 

We totally agreed with you. The estimated prevalence of endometriosis and PCOS were 25-50% and 40% among infertile women. 

Prevalence is based on both incidence and duration of illness. High prevalence of a disease within a population might reflect high incidence or prolonged survival without cure or both. Thus, prevalence rate is always higher than incidence rate [6]. Endometriosis and PCOS were treatable disease [1-5], therefore, it may also be the reason of no cases of endometriosis and PCOS in our cohort of 2400 women with infertility. 

However, we agreed to your precise recommendation as the limitation of our study and have revised these mistakes based on your recommendation in Limitation part. Fourth, there has no incidence of endometriosis or PCOS among infertile women in case group.

References:

1. Küpker W, Felberbaum RE, Krapp M, Schill T, Malik E, Diedrich K. Use of GnRH antagonists in the treatment of endometriosis. Reprod Biomed Online. 2002 Jul-Aug;5(1):12-6. doi: 10.1016/s1472-6483(10)61590-8. PMID: 12470539.

2. Rabinovici J, Stewart EA. New interventional techniques for adenomyosis. Best Pract Res Clin Obstet Gynaecol. 2006 Aug;20(4):617-36. doi: 10.1016/j.bpobgyn.2006.02.002. Epub 2006 Aug 24. PMID: 16934530.

3. Hirsch M, Dhillon-Smith R, Cutner AS, Yap M, Creighton SM. The Prevalence of Endometriosis in Adolescents with Pelvic Pain: A Systematic Review. J Pediatr Adolesc Gynecol. 2020 Dec;33(6):623-630. doi: 10.1016/j.jpag.2020.07.011. Epub 2020 Jul 29. PMID: 32736134.

4. Khadilkar S. S. (2019). Can Polycystic Ovarian Syndrome be cured? Unfolding the Concept of Secondary Polycystic Ovarian Syndrome!. Journal of obstetrics and gynaecology of India, 69(4), 297–302. https://doi.org/10.1007/s13224-019-01253-z

5. Arain F, Arif N, Halepota H. Frequency and outcome of treatment in polycystic ovaries related infertility. Pak J Med Sci. 2015;31(3):694-9. doi: 10.12669/pjms.313.8003. PMID: 26150870; PMCID: PMC4485297.

6. https://www.cdc.gov/csels/dsepd/ss1978/lesson3/section2.html

4. There is no discussion of how they determined the duration of the diagnosis of OSA relative to the timing of the diagnosis of infertility.

Response: Thank you for your thorough review and salient observations. We have added table 4 to determine the duration of the diagnosis of OSA relative to the timing of the diagnosis of infertility.

5. The abstract suggests a different protocol, which would have been to look at women with OSA and see how many are infertile while the actual protocol was to determine odds ratio of having OSA in a cohort of subjects who are infertile.

Response: Thank you for your thorough review and salient observations. We have revised these mistakes based on your recommendation in the abstract. 

To determine the risk of having obstructive sleep apnea in a cohort of female patients who are infertile.

Introduction:

6. There is a discussion about effect of physiological consequences of OSA on sperm production in mice. This does not seem relevant to a paper on female infertility. Consider discussing more about female-relevant hormones, such as you do in the discussion.

Response: Thank you for your thorough review and salient observations. We have revised these mistakes based on your recommendation. We have revised the sentence that related to sperm production in mice to “Scientists have demonstrated young to middle-aged mice which experienced chronic intermittent hypoxia, have reduced the reproductive functions via the decreased of nitric oxide production” and added 4 new references to discuss more about female-relevant hormones in discussion part.

In addition, sex hormone also participated in the distribution of adipose tissue and muscle function in upper respiratory tract [1]. Scientist have demonstrated that high progesterone level in pregnancy status had the protective role in developing OSA, even in obese pregnant women [2, 3]. Greater activity of dilator muscle in upper respiratory tract was also seen in progesterone-dominant luteal phase [4]. 

References:

1. Fotis Kapsimalis, MD, Meir H. Kryger, MD, FRCPC, Gender and Obstructive Sleep Apnea Syndrome, Part 2: Mechanisms, Sleep, Volume 25, Issue 5, August 2002, Pages 497–504, https://doi.org/10.1093/sleep/25.5.497

2. Brownell LG, West P, Kryger MH. Breathing during sleep in normal pregnant women. Am Rev Respir Dis. 1986;133(1):38-41.

3. Maasilta P, Bachour A, Teramo K, Polo O, Laitinen LA. Sleeprelated disordered breathing during pregnancy in obese women. Chest.

2001;120(5):1448-54.

4. Popovic RM, White DP. Upper airway muscle activity in normal women: influence of hormonal status. J Appl Physiol 1998;84(3):1055-1062.

Study Population:

7. Did you exclude women who were infertile because of male infertility issues?

Response: Thank you for your thorough review and salient observations. We extracted data from female infertility which was definitely diagnosed in these 14 years follow up case control study after male infertility has been excluded. 

8. How did you determine the prior exposure of OSA?

Response: Thank you for your thorough review and salient observations. We determined the prior exposure of OSA by the first infertility diagnosis. We have added table 4 to determine the duration of the diagnosis of OSA relative to the timing of the diagnosis of infertility.

9. Last sentence is incomplete.

Response: Thank you for your thorough review and salient observations. We have revised these mistakes based on your recommendation. We have deleted the unnecessary words.

Statistical Analysis

10. There is mention of a Table 5 that is not included in the submission.

Response: Thank you for your thorough review and salient observations. We have revised these mistakes based on your recommendation. We have added table 5 (now table S2) in the end of manuscripts. 

Results:

11. Table 3 is confusing—consider deleting these data. Table 2 is sufficient

Response: Thank you for your thorough review and salient observations. We have deleted Table 3 based on your recommendation. 

Discussion:

12. You did not demonstrate that women with OSA had twice the risk of infertility but, rather, that infertile women had an odds ratio of 2.1 of having OSA compared with women who were fertile.

Response: Thank you for your thorough review and salient observations. We have revised these mistakes based on your recommendation by substitute the sentence to “In this nationwide, population-based, case-control study of over 14 years, we have found that infertile women had an odds ratio of 2.1 of having OSA compared with women who were fertile by creating a 2:1 matched control group with women who were not infertile”, instead of “women with OSA had twice the risk of infertility” (page 12).

13. The fifth paragraph is confusing and the conclusion does not follow. The sex hormones do not mask the severity—they influence the severity.

Response: Thank you for your thorough review and salient observations. We totally agreed with you. We have revised “Sex hormones do not mask the severity” to 

sex hormones may influence the severity of OSA (page 15).

MINOR WEAKNESSES

Suggested Content Edits:

Study Population:

1. “The diagnosis of OSA is made BY polysomnography . . . .”

Response: Thank you for your thorough review and salient observations. We have revised these mistakes based on your recommendation (page 16). 

Statistical Analysis:

2. “Correlation[s] analysis WAS used”

Response: Thank you for your thorough review and salient observations. We have revised these mistakes based on your recommendation (page 10). 

3. “Women aged 26-30 years and 31-35 years had the highest percentage”. I believe you mean “constituted the highest percentage”. If this is not what you mean, then please state what they had the highest percentage of.

Response: Thank you for your thorough review and salient observations. We have revised these mistakes based on your recommendation. 

Women aged 26-30 years and 31-35 years were more than other aged groups in our study population (28.0% and 32.29% respectively).

Discussion,

4. 1st paragraph, last sentence: “This article is the second study that our groups have shown that women . . . infertility”. Consider substituting instead “This article is the second study in which our group has shown that women with sleep disorders . . . infertility.”

Response: Thank you for your thorough review and salient observations. We have revised these mistakes based on your recommendation (page 12). 

5. 3rd paragraph: “. . . elderly women are more likely to become infertile”—I believe you want to say “. . . OLDER women are more likely to BE infertile”.

Response: Thank you for your thorough review and salient observations. We have revised these mistakes based on your recommendation (page 13). 

Endnotes:

6. Endnotes 31 and 32 do not relate to the sentence they are in.

Response: Thank you for your thorough review and salient observations. We have revised these mistakes based on your recommendation. 

We revised the reference to “Potter GD, Skene DJ, Arendt J, Cade JE, Grant PJ, Hardie LJ. Circadian rhythm and sleep disruption: causes, metabolic consequences, and countermeasures. Endocrine reviews 2016; 37(6): 584-608.”

Last, we are deeply honored by the time and effort you spent in reviewing this manuscript. In reviewing and revising our text, we are motivated to read more and thus learn more from your criticisms.

---

## [Decision Letter · Decision Letter 1]

25 Aug 2021

PONE-D-21-15342R1

Obstructive sleep apnea increases risk of female infertility : A 14-year follow-up nationwide population-based cohort study

PLOS ONE

Dear Dr. Chien,

Thank you for submitting your manuscript to PLOS ONE. After careful consideration, we feel that it has merit but does not fully meet PLOS ONE’s publication criteria as it currently stands. Therefore, we invite you to submit a revised version of the manuscript that addresses the points raised during the review process.

While the first reviewer feels that you have responded to her comments in a satisfactory manner, the second reviewer still has concerns at this time. In reading her comments, I must agree that you have two different analyses here and both need to be clearly delineated:

1. You find a higher prevalence of OSA in patients with infertility

2. OSA increases the risk of infertility

I also agree with some of the other changes the second reviewer recommends; they would greatly improve the manuscript and readability.  I strongly urge you to consider these changes. 

We look forward to receiving your revised manuscript.

Kind regards,

James Andrew Rowley

Academic Editor

PLOS ONE

Journal Requirements:

Reviewers' comments:

Reviewer's Responses to Questions

**Comments to the Author**

1. If the authors have adequately addressed your comments raised in a previous round of review and you feel that this manuscript is now acceptable for publication, you may indicate that here to bypass the “Comments to the Author” section, enter your conflict of interest statement in the “Confidential to Editor” section, and submit your "Accept" recommendation.

Reviewer #1: All comments have been addressed

Reviewer #2: (No Response)

2. Is the manuscript technically sound, and do the data support the conclusions?

Reviewer #1: Yes

Reviewer #2: Partly

3. Has the statistical analysis been performed appropriately and rigorously? 

Reviewer #1: Yes

Reviewer #2: I Don't Know

4. Have the authors made all data underlying the findings in their manuscript fully available?

Reviewer #1: Yes

Reviewer #2: Yes

5. Is the manuscript presented in an intelligible fashion and written in standard English?

Reviewer #1: Yes

Reviewer #2: Yes

6. Review Comments to the Author

Reviewer #1: (No Response)

Reviewer #2: Thank you for your work addressing both my concerns and those of the other reviewer. This revision makes it more clear that you are attempting to do two different analyses, which still needs to be clearly enunciated (see comments below). In addition, I am still concerned and confused about the absence of cases of endometriosis and the presence of just one case of PCOS (which happened to be in the fertile cohort). Please address these issues directly, particularly as I found an analysis of the LHID covering a similar period of time that was published in 2019 (https://www.liebertpub.com/doi/full/10.1089/jwh.2018.7648) looking at the relationship of PCOS and periodontal disease that identified 744 patients with PCOS and it is surprising that not a single one of these cases also had a diagnosis of infertility and did not have one of your exclusionary criteria. Finally, I have some issues with your description about the timing of OSA diagnosis and onset of infertility and I believe that the discussion about seasonality is less relevant.

MAJOR WEAKNESSES

ABSTRACT

Objectives: There are actually two objectives: to determine the risk of having OSA in a cohort of female subjects who are infertile and the odds of being infertile in women with OSA.

Results: Highlight the disparity between the prevalence of OSA in the infertile versus fertile group: “Of those women with infertility, 1.38% had a history of OSA compared with 0.63% of fertile controls (p=0.002).

Conclusion: You did not show that OSA is underdiagnosed nor did you fully show that if untreated, it increases the likelihood of infertility. Please be more precise: “Our study showed that OSA is more commonly seen in infertile women and increases the odds that a woman will be infertile. In addition, the likelihood of being diagnosed with infertility increases with time from the diagnosis of OSA [although please see my discussion about this particularly factor below]. More studies need to be done on the whether or not diagnosing and treating OSA can decrease the rate of infertility.”

INTRODUCTION

Third Paragraph: Please delete the whole section about mice as it is irrelevant and relates to male mice.

RESULTS

Second Paragraph: You should have the increased likelihood of subjects with OSA to be infertile at the end of this paragraph where you discuss the other results from Table 2 rather than coming after the information about stratification by possible years of exposure to OSA that starts paragraph 3.

Third Paragraph: I would eliminate the discussion about years of prior OSA exposure since you do not know which patients, if any, were treated nor do you know severity, rendering this a statistically significant result that may or may not have clinical relevance. It is enough to establish an association between having a diagnosis of OSA and one of infertility unless you had more information. Also, Tables 3 and 4 are confusing—it is not clear what you mean about first and last exposure to OSA prior to diagnosis of infertility.

DISCUSSION

First Paragraph: You not only found that women with OSA had an odds ratio of 2.1 of being infertile but also that infertile women were more likely to have OSA.

Third Paragraph: I do not think it adds much and would consider deleting it.

Fourth Paragraph:

1. Is it clear that OSA increases the risk of PCOS or, rather, that patients with PCOS are at greater risk of having OSA, not just because of obesity but also because of hyperadrenergism? Also, I am not aware of OSA increasing the risk of endometriosis. Reference 29 talks about progesterone—I do not think it addresses what you wrote.

2. Reference 32 describes the presentation of sleep-disordered breathing in women and does not discuss the HPA axis and infertility. Either delete this sentence or put in the correct reference. Perhaps you mean to be referencing reference 32 from the initial draft and the article by Ferin?

3. The last 2 sentences seem relevant as this is not a study of salivary cortisol.

Fifth Paragraph:

Consider deleting this. It is not relevant to your argument and has some confusing concepts. For example, the lower prevalence of OSA In women is not because they have different presenting symptoms, it is because of other reasons you mention.

Sixth Paragraph:

I have difficulty with the concept of determining an association of the season of onset of infertility with the date of diagnosis so I would eliminate this entire finding/discussion.

Seventh Paragraph/Limitations:

1. You need to mention that when assessing the duration of OSA, it is approximated by date of diagnosis code and that you do not know whether or not it is treated subsequent to that. I would delete the analysis of this factor because of the limited knowledge but if you keep it in, you need to mention this as a limitation.

2. Secondly, you mention the low rate of PCOS (I think there was 1 case in the fertile group) and endometriosis but do not then provide any explanation and as above, I am concerned given another recent publication that you found so few cases of PCOS overall.

CONCLUSION

You conclude that OSA is more common in infertile women but left out what you had prior that women with OSA have a greater likelihood of being infertile so please add this back. I believe you have greater difficulty with the length of exposure given that you do not know if women are treated or not and should narrow your conclusion to the increased prevalence and likelihood and conclude that infertile women should be screened for signs and symptoms of OSA.

TABLES

Table 2: You do not mention what variables you adjusted for in the adjusted odds ratio. Please add that to the footnote at the bottom of the table. Delete the “without” and “with” headings under the associated disease as they are not necessary and it is confusing.

Tables 3 and 4: I do not understand what is meant by first and last exposure to OSA prior to the diagnosis of infertility. Given how speculative this information is, you might to consider deleting it since you do not know if the OSA is treated or not.

MINOR WEAKNESSES

INTRODUCTION

Second paragraph: 1. website reference to definition of infertility should be a footnote; 2. The reduction in fertility rate could be a function of societal change and not necessarily indicate a change in rate of infertility. Consider rephrasing or eliminating that discussion.

Third paragraph: “Infertility can be associated with multiple factors”.

MATERIALS AND METHODS

Study Population, first paragraph: The diagnosis of OSA is made BY polysomnography.

RESULTS

First Paragraph: Move the sentence about the prevalence of women of different age groups before the sentence about the prevalence of OSA as it interrupts the flow.

DISCUSSION:

6th paragraph: Please edit the sentence about the cows to be grammatically correct.

7. PLOS authors have the option to publish the peer review history of their article (what does this mean?). If published, this will include your full peer review and any attached files.

Reviewer #1: No

Reviewer #2: No

---

## [Author Response · Author response to Decision Letter 1]

7 Oct 2021

Answer to Editor’s and Reviewer's comments

Thank you for your positive comments on this manuscript. The responses to the raised questions are below.

Reviewer #2: Thank you for your work addressing both my concerns and those of the other reviewer. This revision makes it more clear that you are attempting to do two different analyses, which still needs to be clearly enunciated (see comments below). In addition, I am still concerned and confused about the absence of cases of endometriosis and the presence of just one case of PCOS (which happened to be in the fertile cohort). Please address these issues directly, particularly as I found an analysis of the LHID covering a similar period of time that was published in 2019 (https://www.liebertpub.com/doi/full/10.1089/jwh.2018.7648) looking at the relationship of PCOS and periodontal disease that identified 744 patients with PCOS and it is surprising that not a single one of these cases also had a diagnosis of infertility and did not have one of your exclusionary criteria. Finally, I have some issues with your description about the timing of OSA diagnosis and onset of infertility and I believe that the discussion about seasonality is less relevant.

Response: Thank you for your constructive critique. We have revised one by one according to your recommendations below. 

MAJOR WEAKNESSES

ABSTRACT

Objectives: There are actually two objectives: to determine the risk of having OSA in a cohort of female subjects who are infertile and the odds of being infertile in women with OSA.

Results: Highlight the disparity between the prevalence of OSA in the infertile versus fertile group: “Of those women with infertility, 1.38% had a history of OSA compared with 0.63% of fertile controls (p=0.002).

Conclusion: You did not show that OSA is underdiagnosed nor did you fully show that if untreated, it increases the likelihood of infertility. Please be more precise: “Our study showed that OSA is more commonly seen in infertile women and increases the odds that a woman will be infertile. In addition, the likelihood of being diagnosed with infertility increases with time from the diagnosis of OSA [although please see my discussion about this particularly factor below]. More studies need to be done on the whether or not diagnosing and treating OSA can decrease the rate of infertility.”

Response: Thank you for your thorough review and salient observations. We have revised them according to your recommendation. 

Objectives: To determine the risk of having OSA in a cohort of female subjects who are infertile and the odds of being infertile in women with OSA.

Patients and methods: A nationwide, case-control study of female patients 20 years or older diagnosed with female infertility living in Taiwan, from January 1, 2000, through December 31, 2013 (N = 4,078). We identified women who were infertile and created a 2:1 matched control group with women who were not infertile. We used multivariable logistic regression analysis to further estimate the effects of OSA on female infertility.

Results: In this 14- year retrospective study, we included 4,078 patients having an initial diagnosis of female infertility. Of those women with infertility, 1.38% had a history of OSA compared with 0.63% of fertile controls (p=0.002). The mean ages in the study groups were 32.19 ± 6.20 years, whereas the mean ages in the control groups were 32.24 ± 6.37years. Women with OSA had 2.101- times the risk of female infertility compared to women without OSA (p<0.001). 

Conclusion: Our study showed that OSA is more commonly seen in infertile women and increases the odds that a woman will be infertile. More studies need to be done on the whether or not diagnosing and treating OSA can decrease the rate of infertility.

INTRODUCTION

Third Paragraph: Please delete the whole section about mice as it is irrelevant and relates to male mice.

Response: Thank you for your thorough review and salient observations. We have deleted the whole section about mice as it is irrelevant and relates to male mice.

Infertility can be associated with multiple factors, such as inflammation, obesity, intermittent hypoxia and sympathetic activation8,9. In addition, continuous positive airway pressure treatment showed the increased rate of successful intercourse attempts and increased of the score in International Index of Erectile Function 10,11. In our previous study, we have found an increased risk of infertility in female patients with non-apnea sleep disorder 12. However, little is known about whether OSA is associated with a risk of female infertility. 

References:

8. Torres M, Laguna-Barraza R, Dalmases M, et al. Male fertility is reduced by chronic intermittent hypoxia mimicking sleep apnea in mice. Sleep 2014; 37(11): 1757-65.

9. Soukhova-O'Hare GK, Shah ZA, Lei Z, Nozdrachev AD, Rao CV, Gozal D. Erectile dysfunction in a murine model of sleep apnea. American journal of respiratory and critical care medicine 2008; 178(6): 644-50.

10. Li X, Dong Z, Wan Y, Wang Z. Sildenafil versus continuous positive airway pressure for erectile dysfunction in men with obstructive sleep apnea: a meta-analysis. The Aging Male 2010; 13(2): 82-6.

11. Perimenis P, Karkoulias K, Konstantinopoulos A, et al. Sildenafil versus continuous positive airway pressure for erectile dysfunction in men with obstructive sleep apnea: a comparative study of their efficacy and safety and the patient's satisfaction with treatment. Asian journal of andrology 2007; 9(2): 259-64.

12. Wang ID, Liu YL, Peng CK, et al. Non-apnea sleep disorder increases the risk of subsequent female infertility-a nationwide population-based cohort study. Sleep 2018; 41(1): zsx186.

RESULTS

Second Paragraph: You should have the increased likelihood of subjects with OSA to be infertile at the end of this paragraph where you discuss the other results from Table 2 rather than coming after the information about stratification by possible years of exposure to OSA that starts paragraph 3.

Response: Thank you for your thorough review and salient observations. We have revised according to your recommendation. 

Multivariable logistic regression analysis was used in this retrospective study. We listed factors for female infertility in Table 2. There were no significant differences in the risk of female infertility between patients with and without gynecological disorders, endocrine disorders, or concomitant comorbidities, including hypertension, diabetes mellitus, hyperlipidemia, COPD, chronic kidney disease, coronary heart disease, stroke, obesity, anxiety, and depression (Table 2). The increased likelihood of subjects with OSA to be infertile was also showed in Table 2 (adjusted odds ratio, 2.101; p<0.001). 

Third Paragraph: I would eliminate the discussion about years of prior OSA exposure since you do not know which patients, if any, were treated nor do you know severity, rendering this a statistically significant result that may or may not have clinical relevance. It is enough to establish an association between having a diagnosis of OSA and one of infertility unless you had more information. Also, Tables 3 and 4 are confusing—it is not clear what you mean about first and last exposure to OSA prior to diagnosis of infertility.

Response: Thank you for your thorough review and salient observations. We have deleted them according to your recommendation.

DISCUSSION

First Paragraph: You not only found that women with OSA had an odds ratio of 2.1 of being infertile but also that infertile women were more likely to have OSA.

Response: Thank you for your thorough review and salient observations. We have revised according to your recommendation. Our study is the largest retrospective study to date demonstrating the association of female infertility with OSA. In this nationwide, population-based, case-control study of over 14 years, we have found that infertile women had an odds ratio of 2.1 of having OSA compared with women who were fertile but also infertile women were more likely to have OSA. This article is the second study in which our groups has shown that women with sleep disorders could link to female infertility. In our previous article, we have found that women with non-apnea sleep disorder had a 3.718-fold risk of female infertility compared with the control cohort12.

Reference:

12. Wang ID, Liu YL, Peng CK, et al. Non-apnea sleep disorder increases the risk of subsequent female infertility-a nationwide population-based cohort study. Sleep 2018; 41(1): zsx186.

Third Paragraph: I do not think it adds much and would consider deleting it.

Response: Thank you for your thorough review and salient observations. We have deleted third paragraph according to your recommendation. 

Fourth Paragraph:

1. Is it clear that OSA increases the risk of PCOS or, rather, that patients with PCOS are at greater risk of having OSA, not just because of obesity but also because of hyperadrenergism? Also, I am not aware of OSA increasing the risk of endometriosis. Reference 29 talks about progesterone—I do not think it addresses what you wrote.

Response: Thank you for your thorough review and salient observations. PCOS was defined as 1) oligoovulation, 2) clinical hyperandrogenism (i.e. hirsutism) and/or hyperandrogenemia, and 3) exclusion of other related disorders, such as hyperprolactinemia, thyroid abnormalities, and non-classic adrenal hyperplasia1. Therefore, PCOS are at greater risk of having OSA, not just because of obesity but also because of hyperadrenergism. Also, due to OSA did not increase the risk of endometriosis, we have deleted the word “endometriosis” according to your suggestion.

References:

1. Knochenhauer, E. S., Key, T. J., Kahsar-Miller, M., Waggoner, W., Boots, L. R., & Azziz, R. (1998). Prevalence of the polycystic ovary syndrome in unselected black and white women of the southeastern United States: a prospective study. The Journal of clinical endocrinology and metabolism, 83(9), 3078–3082. https://doi.org/10.1210/jcem.83.9.5090

2. Reference 32 describes the presentation of sleep-disordered breathing in women and does not discuss the HPA axis and infertility. Either delete this sentence or put in the correct reference. Perhaps you mean to be referencing reference 32 from the initial draft and the article by Ferin?

Response: 

Thank you for your thorough review and salient observations. We have corrected the reference to Ferin M. Stress and the Reproductive Cycle. The Journal of Clinical Endocrinology & Metabolism 1999; 84(6): 1768-74.

3. The last 2 sentences seem relevant as this is not a study of salivary cortisol.

Response: Thank you for your thorough review and salient observations. We have revised the reference according to your suggestion.

Patients with OSA are at risk of metabolic disorders, including an irregular menstrual cycle, obesity and PCOS29. The menstrual cycle is modulated by the hypothalamus–pituitary–gonadal axis. The hypothalamus releases gonadotropin-releasing hormone, the pituitary gland produces follicle-stimulating hormone and luteinizing hormone, and the ovary produces estrogen and testosterone. Kloss et al. hypothesized that sleep disorders may activate the hypothalamus–pituitary–gonadal axis and alter sex hormones in the follicular, ovulation, luteal, and menstruation phases, ultimately resulting in inferility30. In addition, the interruption of breathing during sleep as a result of OSA can cause circadian dysrhythmia due to the increased levels of melatonin and cortisol31. The increased of cortisol levels downregulated the hypothalamic-pituitary-adrenal axis and involved in the inhibition of GnRH at the pituitary level, which altered sex hormone profile and thus leading to infertility32. Moreover, women with higher salivary cortisol levels have lower levels of estradiol during the follicular and luteal phases, accompanied by an increased concentration of cortisol and estrogen receptors in the mood regulation region of the brain33. Therefore, salivary cortisol levels are inversely proportional to estradiol levels, which may lead to missed or irregular periods and make getting pregnant difficult.

References:

29. Nitsche K, Ehrmann DA. Obstructive sleep apnea and metabolic dysfunction in polycystic ovary syndrome. Best Pract Res Clin Endocrinol Metab 2010; 24(5): 717-30.

30. Kloss JD, Perlis ML, Zamzow JA, Culnan EJ, Gracia CR. Sleep, sleep disturbance, and fertility in women. Sleep Med Rev 2015; 22: 78-87.

31. Potter GDM, Skene DJ, Arendt J, Cade JE, Grant PJ, Hardie LJ. Circadian Rhythm and Sleep Disruption: Causes, Metabolic Consequences, and Countermeasures. Endocr Rev 2016; 37(6): 584-608.

32. Ferin M. Stress and the Reproductive Cycle. The Journal of Clinical Endocrinology & Metabolism 1999; 84(6): 1768-74.

33. Hamidovic A, Karapetyan K, Serdarevic F, Choi SH, Eisenlohr-Moul T, Pinna G. Higher Circulating Cortisol in the Follicular vs. Luteal Phase of the Menstrual Cycle: A Meta-Analysis. Front Endocrinol (Lausanne) 2020; 11: 311-.

Fifth Paragraph:

Consider deleting this. It is not relevant to your argument and has some confusing concepts. For example, the lower prevalence of OSA In women is not because they have different presenting symptoms, it is because of other reasons you mention.

Response: Thank you for your thorough review and salient observations. We have deleted it according to your recommendation. 

Sixth Paragraph:

I have difficulty with the concept of determining an association of the season of onset of infertility with the date of diagnosis so I would eliminate this entire finding/discussion.

Response: Thank you for your thorough review and salient observations. We have deleted it according to your recommendation. 

Seventh Paragraph/Limitations:

1. You need to mention that when assessing the duration of OSA, it is approximated by date of diagnosis code and that you do not know whether or not it is treated subsequent to that. I would delete the analysis of this factor because of the limited knowledge but if you keep it in, you need to mention this as a limitation.

Response: Thank you for your thorough review and salient observations. We have deleted the analysis of this factor and added it as limitation, according to your recommendation. 

This study had some limitations. First, our database could not provide detailed information about laboratory data or about the psychological impact of infertility and the use of assisted reproductive technologies. Second, although the diagnosis of OSA is made by polysomnography, we could not obtain the severity of OSA due to the de-identification in the database. Therefore, studies regarding the severity of OSA and subsequent female infertility are warranted. Third, although the study draws subjects from a large database, however, the number of infertility and OSA was only 33 and the number of OSA with no infertility in the control group was only 30 subjects. Fourth, there has no incidence of endometriosis or PCOS among infertile women in case group. A study has showed no infertility case in polycystic ovaries after performing blood hormones test and gynecologic ultrasonography and no constructive study that shows the relationship between OSA and endometriosis44. Fifth, the duration of OSA is approximated by date of diagnosis code but we do not know whether or not it is treated subsequent to that. Despite the listed limitations, our study provided a large group of patients and its longitudinal effects of over 14 years.

Reference:

44. Tong, C., Wang, Y. H., Yu, H. C., & Chang, Y. C. (2019). Increased Risk of Polycystic Ovary Syndrome in Taiwanese Women with Chronic Periodontitis: A Nationwide Population-Based Retrospective Cohort Study. Journal of women's health (2002), 28(10), 1436–1441. https://doi.org/10.1089/jwh.2018.7648

2. Secondly, you mention the low rate of PCOS (I think there was 1 case in the fertile group) and endometriosis but do not then provide any explanation and as above, I am concerned given another recent publication that you found so few cases of PCOS overall.

Response: Thank you for your thorough review and salient observations. We have advised it according to your recommendation. 

This study had some limitations. First, our database could not provide detailed information about laboratory data or about the psychological impact of infertility and the use of assisted reproductive technologies. Second, although the diagnosis of OSA is made by polysomnography, we could not obtain the severity of OSA due to the de-identification in the database. Therefore, studies regarding the severity of OSA and subsequent female infertility are warranted. Third, although the study draws subjects from a large database, however, the number of infertility and OSA was only 33 and the number of OSA with no infertility in the control group was only 30 subjects. Fourth, there has no incidence of endometriosis or PCOS among infertile women in case group. A study has showed no infertility case in polycystic ovaries after performing blood hormones test and gynecologic ultrasonography and no constructive study that shows the relationship between OSA and endometriosis44. Fifth, the duration of OSA is approximated by date of diagnosis code but we do not know whether or not it is treated subsequent to that. Despite the listed limitations, our study provided a large group of patients and its longitudinal effects of over 14 years.

Reference:

44. Tong, C., Wang, Y. H., Yu, H. C., & Chang, Y. C. (2019). Increased Risk of Polycystic Ovary Syndrome in Taiwanese Women with Chronic Periodontitis: A Nationwide Population-Based Retrospective Cohort Study. Journal of women's health (2002), 28(10), 1436–1441. https://doi.org/10.1089/jwh.2018.7648

CONCLUSION

You conclude that OSA is more common in infertile women but left out what you had prior that women with OSA have a greater likelihood of being infertile so please add this back. I believe you have greater difficulty with the length of exposure given that you do not know if women are treated or not and should narrow your conclusion to the increased prevalence and likelihood and conclude that infertile women should be screened for signs and symptoms of OSA.

Response: Thank you for your thorough review and salient observations. 

In conclusion, this study showed that OSA is more commonly seen in infertile women and increases the odds that a woman will be infertile. Therefore, infertile women should be screened for signs and symptoms of OSA, which may help to increase female fertility rate.

TABLES

Table 2: You do not mention what variables you adjusted for in the adjusted odds ratio. Please add that to the footnote at the bottom of the table. Delete the “without” and “with” headings under the associated disease as they are not necessary and it is confusing.

Response: Thank you for your thorough review and salient observations. We have revised them in Material and methods and Table section according to your recommendation.

We compared the study and control groups with regard to characteristics and common comorbidities, including hypertension, diabetes mellitus, hyperlipidemia, and COPD, by using chi-squared tests. The mean ages of the two groups were compared using Student’s t-test. The odds ratio (OR) for factors potentially associated with female infertility were evaluated using multivariable logistic regression with and without stratification. The variables that adjusted in the odds ratio were the variables that listed in the table. Correlation analysis was used to study the strength of a relationship between age groups and the variables listed in Table S2. All comparisons were two-tailed, and p-values <0.05 were considered statistically significant. The statistical analyses were performed using IBM SPSS v 22.0 software.

Tables 3 and 4: I do not understand what is meant by first and last exposure to OSA prior to the diagnosis of infertility. Given how speculative this information is, you might to consider deleting it since you do not know if the OSA is treated or not.

Response: Thank you for your thorough review and salient observations. We have deleted Table 3 and 4 according to your recommendation. 

MINOR WEAKNESSES

INTRODUCTION

Second paragraph: 1. website reference to definition of infertility should be a footnote; 2. The reduction in fertility rate could be a function of societal change and not necessarily indicate a change in rate of infertility. Consider rephrasing or eliminating that discussion.

Response: Thank you for your thorough review and salient observations. We have revised them according to your recommendation.

According to the American College of Obstetricians and Gynecologists (ACOG), the definition of infertility is the failure to conceive after 1 year or more of regular unprotected sexual intercourse6. The prevalence of infertility has increased since 1990; in 2010, approximately 48.5 million individuals worldwide suffered from infertility7. Moreover, according to the data published by Taiwan’s Ministry of the Interior, the fertility rates of women within childbearing age decreased from 1.680 births per woman in 2000 to 1.080 births per woman in 2018, with the mean age of the women at their first birth increasing from 22.88 years to 30.90 years during that period. Female infertility can result from various conditions, including endometriosis, pelvic adhesion, polycystic ovary syndrome, tubal blockage, hyperprolactinemia, and congenital or acquired uterine or ovarian abnormalities8.

References:

6. Medicine CoGPftASfR, Obstetricians tACo, Gynecologists. Infertility workup for the Women’s Health Specialist. Obstet Gynecol 2019; 133: e377-e84.

7. Mascarenhas MN, Flaxman SR, Boerma T, Vanderpoel S, Stevens GA. National, regional, and global trends in infertility prevalence since 1990: a systematic analysis of 277 health surveys. PLoS Med 2012; 9(12): e1001356.

8. Abrao MS, Muzii L, Marana R. Anatomical causes of female infertility and their management. International journal of gynaecology and obstetrics: the official organ of the International Federation of Gynaecology and Obstetrics 2013; 123 Suppl 2: S18-S24.

Third paragraph: “Infertility can be associated with multiple factors”.

Response: Thank you for your thorough review and salient observations. We have revised it according to your recommendation. 

MATERIALS AND METHODS

Study Population, first paragraph: The diagnosis of OSA is made BY polysomnography.

Response: Thank you for your thorough review and salient observations. We have revised it according to your recommendation. 

RESULTS

First Paragraph: Move the sentence about the prevalence of women of different age groups before the sentence about the prevalence of OSA as it interrupts the flow.

Response: Thank you for your thorough review and salient observations. We have revised it according to your recommendation. 

We identified 4,078 female patients in the database who had been diagnosed with infertility by the end of the study. After applying the exclusion criteria, 2,400 patients were included and assigned as the study group in the analysis. The control group comprised 4,800 matched women without infertility. The mean ages in the study groups were 32.19 ± 6.20 years, whereas the mean ages in the control groups were 32.24 ± 6.37 years. Table 1 summarizes the characteristics of the study and the control groups. There were significantly more patients who had been diagnosed with OSA in the study group than in the control group (1.38% vs. 0.63%; p = .002). In addition, we have found that 33 patients in the study group and 30 patients in the reference group, who have had prior exposure to OSA. Women aged 26-30 years and 31-35 years were more than other aged groups in our study population (28.0% and 32.29% respectively).

DISCUSSION:

6th paragraph: Please edit the sentence about the cows to be grammatically correct.

Response: Thank you for your thorough review and salient observations. We have deleted the paragraph because the discussion about seasonality is less relevant.

Last, we are deeply honored by the time and effort you spent in reviewing this manuscript. In reviewing and revising our text, we are motivated to read more and thus learn more from your criticisms.

---

## [Decision Letter · Decision Letter 2]

27 Oct 2021

PONE-D-21-15342R2Obstructive sleep apnea increases risk of female infertility : A 14-year follow-up nationwide population-based cohort studyPLOS ONE

Dear Dr. Chien,

Thank you for submitting your manuscript to PLOS ONE. After careful consideration, we feel that it has merit but does not fully meet PLOS ONE’s publication criteria as it currently stands. Therefore, we invite you to submit a revised version of the manuscript that addresses the points raised during the review process.

Please make the suggested revisions for Reviewer #2.  

We look forward to receiving your revised manuscript.

Kind regards,

James Andrew Rowley

Academic Editor

PLOS ONE

Journal Requirements:

Reviewers' comments:

Reviewer's Responses to Questions

**Comments to the Author**

1. If the authors have adequately addressed your comments raised in a previous round of review and you feel that this manuscript is now acceptable for publication, you may indicate that here to bypass the “Comments to the Author” section, enter your conflict of interest statement in the “Confidential to Editor” section, and submit your "Accept" recommendation.

Reviewer #2: (No Response)

2. Is the manuscript technically sound, and do the data support the conclusions?

Reviewer #2: Yes

3. Has the statistical analysis been performed appropriately and rigorously? 

Reviewer #2: Yes

4. Have the authors made all data underlying the findings in their manuscript fully available?

Reviewer #2: Yes

5. Is the manuscript presented in an intelligible fashion and written in standard English?

Reviewer #2: Yes

6. Review Comments to the Author

Reviewer #2: Thank you for your attention to my previous comments. This paper is much improved. There are still some minor weaknesses to address but we are almost there. Please see remaining comments below.

Minor Weaknesses

Abstract: delete the sentence in the conclusion about the likelihood of being infertile increasing with time from diagnosis of OSA as you deleted this analysis because of insufficient information.

Introduction, 3rd paragraph: please delete the sentence about CPAP improving erectile function—not relevant to a population of women (and be careful about adjusting your references).

Discussion, 1st paragraph, second sentence: “ . . . found that infertile women had” should be “infertile women have” and 3rd sentence: . . . which our groups have shown” or “which our group has shown”—whichever version is correct.

Discussion, 3rd paragraph: As mentioned before, the first sentence is incorrect. Women with PCOS are at increased risk for OSA; women with OSA are NOT known to be at increased risk for PCOS. Your topic sentence really should be about a theoretical relationship of hormonal dysregulation from OSA leading to infertility. Please change it.

Discussion, 3rd paragraph: . . . “ . . . cortisol31. The increase in cortisol levels can downregulate the HPA axis and inhibit GnRH at the pituitary level, which may alter sex hormone profiles and thus lead to infertility.32”

Discussion, 3rd paragraph: delete the discussion about salivary cortisol. It does not add anything and is confusing.

Discussion 4th paragraph: “ . . . prevalence of OSA In women is lower than IN men”.

Discussion 4th paragraph: the lower prevalence of OSA in pre-menopausal women is not a function of atypical presentation, it simply is. The atypical presentation explains the underdiagnosis of women but the prevalence is still lower in women so please revise this section.

Limitations: Delete the first limitation.

Limitations: fourth limitation: please point out that the absence of women with endometriosis and PCOS in the infertile cohort potentially limits the applicability in populations where these disorders are more common. Please delete the sentence about no infertility in PCOS and reference 45—not the point here.

Limitations: please delete the fifth limitation as you have removed that analysis from this paper.

7. PLOS authors have the option to publish the peer review history of their article (what does this mean?). If published, this will include your full peer review and any attached files.

Reviewer #2: No

---

## [Author Response · Author response to Decision Letter 2]

3 Nov 2021

Answer to Editor’s and Reviewer's comments

Thank you for your positive comments on this manuscript. The responses to the raised questions are below.

Reviewer's Responses to Questions

Comments to the Author

1. If the authors have adequately addressed your comments raised in a previous round of review and you feel that this manuscript is now acceptable for publication, you may indicate that here to bypass the “Comments to the Author” section, enter your conflict of interest statement in the “Confidential to Editor” section, and submit your "Accept" recommendation.

Reviewer #2: (No Response)

2. Is the manuscript technically sound, and do the data support the conclusions?

Reviewer #2: Yes

Response: Thank you for your thorough review and salient observations.

3. Has the statistical analysis been performed appropriately and rigorously?

Reviewer #2: Yes

Response: Thank you for your thorough review and salient observations.

4. Have the authors made all data underlying the findings in their manuscript fully available?

Reviewer #2: Yes

Response: Thank you for your thorough review and salient observations.

5. Is the manuscript presented in an intelligible fashion and written in standard English?

Reviewer #2: Yes

Response: Thank you for your thorough review and salient observations.

6. Review Comments to the Author

Reviewer #2: Thank you for your attention to my previous comments. This paper is much improved. There are still some minor weaknesses to address but we are almost there. Please see remaining comments below.

Response: Thank you for your thorough review and salient observations. We have revised these mistakes according to your suggestion. 

Minor Weaknesses

Abstract: delete the sentence in the conclusion about the likelihood of being infertile increasing with time from diagnosis of OSA as you deleted this analysis because of insufficient information.

Response: Thank you for your thorough review and salient observations. We have revised these mistakes according to your suggestion.

Conclusion: Our study showed OSA is more commonly seen in infertile women, especially those with concomitant comorbidities, but yet its risk is often underdiagnosed. Therefore, early detection of OSA in women, especially in those with atypical symptoms, may help to increase female fertility rate.

Introduction, 3rd paragraph: please delete the sentence about CPAP improving erectile function—not relevant to a population of women (and be careful about adjusting your references).

Response: Thank you for your thorough review and salient observations. We have revised these mistakes according to your suggestion.

Infertility was associated with multiple factors, such as inflammation, obesity, intermittent hypoxia and sympathetic activation. Scientists have demonstrated young to middle-aged mice which experienced chronic intermittent hypoxia, have reduced the reproductive functions via the decreased of nitric oxide production. They have found the association of male infertility in these OSA mice8,9. In our previous study, we have found an increased risk of infertility in female patients with non-apnea sleep disorder10. However, little is known about whether OSA is associated with a risk of female infertility.

References:

8. Torres M, Laguna-Barraza R, Dalmases M, et al. Male fertility is reduced by chronic intermittent hypoxia mimicking sleep apnea in mice. Sleep 2014; 37(11): 1757-65.

9. Soukhova-O'Hare GK, Shah ZA, Lei Z, Nozdrachev AD, Rao CV, Gozal D. Erectile dysfunction in a murine model of sleep apnea. American journal of respiratory and critical care medicine 2008; 178(6): 644-50.

10. Wang ID, Liu YL, Peng CK, et al. Non-apnea sleep disorder increases the risk of subsequent female infertility-a nationwide population-based cohort study. Sleep 2018; 41(1): zsx186.

Discussion, 1st paragraph, second sentence: “ . . . found that infertile women had” should be “infertile women have” and 3rd sentence: . . . which our groups have shown” or “which our group has shown”—whichever version is correct.

Response: Thank you for your thorough review and salient observations. We have revised these mistakes according to your suggestion. 

Our study is the largest retrospective study to date demonstrating the association of female infertility with OSA. In this nationwide, population-based, case-control study of over 14 years, we have found that infertile women had an odds ratio of 2.1 of having OSA compared with women who were fertile. This article is the second study in which our groups has shown that women with sleep disorders could link to female infertility. In our previous article, we have found that women with non-apnea sleep disorder have a 3.718-fold risk of female infertility compared with the control cohort10.

Reference:

10. Wang ID, Liu YL, Peng CK, et al. Non-apnea sleep disorder increases the risk of subsequent female infertility-a nationwide population-based cohort study. Sleep 2018; 41(1): zsx186.

Discussion, 3rd paragraph: As mentioned before, the first sentence is incorrect. Women with PCOS are at increased risk for OSA; women with OSA are NOT known to be at increased risk for PCOS. Your topic sentence really should be about a theoretical relationship of hormonal dysregulation from OSA leading to infertility. Please change it.

Response: Thank you for your thorough review and salient observations. We have revised these mistakes according to your suggestion. 

Patients with OSA are at risk of metabolic disorders, including an irregular menstrual cycle, endometriosis, and uterine leiomyoma27.

Reference:

27. Skatrud J, Dempsey J, Kaiser D. Ventilatory response to medroxyprogesterone acetate in normal subjects: time course and mechanism. J Appl Physiol 1978; 44(6): 939-44.

Discussion, 3rd paragraph: . . . “ . . . cortisol31. The increase in cortisol levels can downregulate the HPA axis and inhibit GnRH at the pituitary level, which may alter sex hormone profiles and thus lead to infertility.32”

Response: Thank you for your thorough review and salient observations. We have revised these mistakes according to your suggestion. 

Discussion, 3rd paragraph: delete the discussion about salivary cortisol. It does not add anything and is confusing.

Response: Thank you for your thorough review and salient observations. We have deleted the discussion about salivary cortisol according to your suggestion. 

Discussion 4th paragraph: “ . . . prevalence of OSA In women is lower than IN men”.

Response: Thank you for your thorough review and salient observations. We have revised these mistakes according to your suggestion. 

Discussion 4th paragraph: the lower prevalence of OSA in pre-menopausal women is not a function of atypical presentation, it simply is. The atypical presentation explains the underdiagnosis of women but the prevalence is still lower in women so please revise this section.

Response: Thank you for your thorough review and salient observations. We have revised these mistakes according to your suggestion. 

It is because of atypical OSA symptoms such as depression, headache, anxiety and insomnia are more frequent presented in women31.

Reference:

31. Collop NA, Adkins D, Phillips BA. Gender differences in sleep and sleep-disordered breathing. Clin Chest Med 2004; 25(2): 257-68.

Limitations: Delete the first limitation.

Response: Thank you for your thorough review and salient observations. We have revised these mistakes according to your suggestion. 

Limitations: fourth limitation: please point out that the absence of women with endometriosis and PCOS in the infertile cohort potentially limits the applicability in populations where these disorders are more common. Please delete the sentence about no infertility in PCOS and reference 45—not the point here.

Response: Thank you for your thorough review and salient observations. We have revised these mistakes according to your suggestion. 

Limitations: please delete the fifth limitation as you have removed that analysis from this paper.

Response: Thank you for your thorough review and salient observations. We have revised these mistakes according to your suggestion. 

7. PLOS authors have the option to publish the peer review history of their article (what does this mean?). If published, this will include your full peer review and any attached files.

Do you want your identity to be public for this peer review? For information about this choice, including consent withdrawal, please see our Privacy Policy.

Reviewer #2: No

Response: Thank you for your thorough review and salient observations.

Last, we are deeply honored by the time and effort you spent in reviewing this manuscript. In reviewing and revising our text, we are motivated to read more and thus learn more from your criticisms.

---

## [Editor Report · Decision Letter 3]

16 Nov 2021

PONE-D-21-15342R3Obstructive sleep apnea increases risk of female infertility : A 14-year follow-up nationwide population-based cohort studyPLOS ONE

Dear Dr. Chien,

Thank you for submitting your manuscript to PLOS ONE. After careful consideration, we feel that it has merit but does not fully meet PLOS ONE’s publication criteria as it currently stands. Therefore, we invite you to submit a revised version of the manuscript that addresses the points raised during the review process.

The academic editor went through the latest revisions for the reviewer and most of have been changed in a satisfactory manner. However, there is still one small change to make, regarding the comment on prevalence (5th paragraph of discussion). The reviewer was pointing out that the lower prevalence in women is NOT due to an atypical presentation as epidemiologic studies, in which all subjects are studied, is lower in women.  However, what is lower is underevaluation because of atypical presentations. Thus,the second sentence of this paragraph should be changed to: 'There is also likely underdiagnosis of OSA in women due to atypical symptoms such as...'. Please submit your revised manuscript by Dec 31 2021 11:59PM. If you will need more time than this to complete your revisions, please reply to this message or contact the journal office at plosone@plos.org. Please include the following items when submitting your revised manuscript:A rebuttal letter that responds to each point raised by the academic editor and reviewer(s). You should upload this letter as a separate file labeled 'Response to Reviewers'.A marked-up copy of your manuscript that highlights changes made to the original version. You should upload this as a separate file labeled 'Revised Manuscript with Track Changes'.An unmarked version of your revised paper without tracked changes. You should upload this as a separate file labeled 'Manuscript'.If applicable, we recommend that you deposit your laboratory protocols in protocols.io to enhance the reproducibility of your results. Protocols.io assigns your protocol its own identifier (DOI) so that it can be cited independently in the future. For instructions see: https://journals.plos.org/plosone/s/submission-guidelines#loc-laboratory-protocols. Additionally, PLOS ONE offers an option for publishing peer-reviewed Lab Protocol articles, which describe protocols hosted on protocols.io. Read more information on sharing protocols at https://plos.org/protocols?utm_medium=editorial-email&utm_source=authorletters&utm_campaign=protocols.

We look forward to receiving your revised manuscript.

Kind regards,

James Andrew Rowley

Academic Editor

PLOS ONE
---

## [Author Response · Author response to Decision Letter 3]

16 Nov 2021

Answer to Editor’s and Reviewer's comments

Thank you for your positive comments on this manuscript. The responses to the raised questions are below.

Reviewer's Responses to Questions

Comments to the Author

The academic editor went through the latest revisions for the reviewer and most of have been changed in a satisfactory manner. However, there is still one small change to make, regarding the comment on prevalence (5th paragraph of discussion). The reviewer was pointing out that the lower prevalence in women is NOT due to an atypical presentation as epidemiologic studies, in which all subjects are studied, is lower in women. However, what is lower is underevaluation because of atypical presentations. Thus, the second sentence of this paragraph should be changed to: 'There is also likely underdiagnosis of OSA in women due to atypical symptoms such as...'.

Response: Thank you for your thorough review and salient observations. We have revised these mistakes according to your suggestion.

Scientists have demonstrated the prevalence of OSA in women is lower than in men2. There is also likely underdiagnosis of OSA in women due to atypical symptoms such as depression, headache, anxiety and insomnia are more frequent presented in women31.

References:

2. Young T, Palta M, Dempsey J, Skatrud J, Weber S, Badr S. The occurrence of sleep-disordered breathing among middle-aged adults. The New England journal of medicine 1993; 328(17): 1230-5.

31. Ferin M. Stress and the Reproductive Cycle. The Journal of Clinical Endocrinology & Metabolism 1999; 84(6): 1768-74.

Last, we are deeply honored by the time and effort you spent in reviewing this manuscript. In reviewing and revising our text, we are motivated to read more and thus learn more from your criticisms.

---

## [Editor Report · Decision Letter 4]

18 Nov 2021

Obstructive sleep apnea increases risk of female infertility : A 14-year follow-up nationwide population-based cohort study

PONE-D-21-15342R4

Dear Dr. Chien,

We’re pleased to inform you that your manuscript has been judged scientifically suitable for publication and will be formally accepted for publication once it meets all outstanding technical requirements.

Kind regards,

James Andrew Rowley

Academic Editor

PLOS ONE
---

## [Editor Report · Acceptance letter]

6 Dec 2021

PONE-D-21-15342R4 

Obstructive sleep apnea increases risk of female infertility: A 14-year nationwide population-based study 

Dear Dr. Chien:

I'm pleased to inform you that your manuscript has been deemed suitable for publication in PLOS ONE. Congratulations! Your manuscript is now with our production department. 

Kind regards, 

on behalf of

Dr. James Andrew Rowley 

Academic Editor

PLOS ONE